# Tellegacy: An Intergenerational Wellness and Health Promotion Project to Reduce Social Isolation and Loneliness in Older Adults: A Feasibility Study

**DOI:** 10.3390/ijerph20237094

**Published:** 2023-11-22

**Authors:** Jeremy Holloway, Owais Sayeed, Donald Jurivich

**Affiliations:** 1Department of Geriatrics, University of North Dakota School of Medicine and Health Sciences, Grand Forks, ND 58202, USA; donald.jurivich@und.edu; 2School of Behavioral and Brain Sciences, University of Texas at Dallas, Richardson, TX 75080, USA; owais.sayeed@utdallas.edu

**Keywords:** social isolation, loneliness, wellness, health promotion, intergenerational connection, older adults

## Abstract

Emerging research demonstrates that social isolation and loneliness are linked to significant physical and mental health conditions. To address these concerns, the Tellegacy program was developed as an intergenerational health-promoting intervention to ameliorate older adult social isolation and loneliness in an effort to increase wellness. The purpose of this study was to reflect on testing of the Tellegacy program as a behavioral intervention. University students trained in goal setting, mindfulness, and listening strategies were paired with 11 older adults in the northern Midwest area via weekly in-person and phone conversations. Oral reminiscence therapies were used and books containing their stories were given to the older adults after participation. Older adults were surveyed using the University of California Los Angeles Loneliness Scale, Satisfaction of Life Scale, and patient health questionnaire-9 (PHQ-9) scale to elucidate the effectiveness of the intervention. Improved scores in loneliness, satisfaction of life, and PHQ-9 demonstrated favorable improvements in older adults. Additional benefits for the student Legacy Builder were revealed from self-reported changes. This suggests the potential benefits of structured encounters between trained students and isolated or lonely older adults. The Tellegacy intergenerational feasibility program warrants further studies to fully demonstrate its impact on health outcomes.

## 1. Introduction

Social isolation and loneliness have emerged as pressing public health concerns within an aging society. This multifaceted issue can be attributed to a range of factors, including the diminishing prevalence of intergenerational living arrangements and increased social and geographic mobility [1]. Simultaneously, ageism and a lack of older adult experiential training among University students in healthcare fields further increase issues of social isolation and loneliness among older adult patients.

Although social isolation and loneliness are used interchangeably, they represent two different conditions. According to the American Association of Retired Persons, “loneliness is the distressing feeling of being alone or separated while social isolation is the lack of social contacts and having few people to interact with regularly” [2]. Absent emotional connectivity with other humans serves as a driving factor for social isolation [3]. Prior research studies highlight how trends in social isolation disproportionally affect the unmarried, older adults, males, and individuals in lower social-class [4].

Nearly one third of adults aged 45 years or older report feeling lonely, while one fourth of adults aged 65 years and older are considered socially isolated [5]. Over the last few years, the incidence of social isolation and loneliness has exacerbated across the United States, which is a problem further compounded by the recent COVID-19 pandemic [6,7]. These issues lead toa further decline in older adult social wellness and physical health [2]. 

Accelerated cardiovascular disease, diabetes, dementia, anxiety, depression, and suicide strongly correlate with these conditions and raise the possibility of rapid aging [8,9,10]. Social isolation significantly increases a person’s risk of premature death from all causes, a risk that may rival that of smoking, obesity, physical inactivity, and frailty [11]. A recent study revealed that poor social relationships characterized by social isolation or loneliness have a 29% increased risk of heart disease and a 32% increased risk of stroke [1]. Furthermore, isolation and loneliness increase the risk of early death by 26%, which equates to the same risk as smoking 15 cigarettes a day [12,13]. 

The current culture of healthcare, with increased rates of burnout and lower rates of employee retention, leads medical staff to carry a negative stigma against older adults, known as ageism. Ageism refers to a combination of anxiety about aging, the knowledge of older stereotypes, and the divergence of older adults in the comparison of the cultural norms from their own upbringings to modern-day norms [14]. Due to this difference, older adults can be discriminated against and treated unequally in comparison to younger patients or individuals within healthcare settings. This discrimination and lack of adequate care results in familial caregivers and other healthcare staff to be overworked when providing care for older adults [15]. The decreased amount of training, quality of healthcare available for older adults, and ageist attitudes contribute to an overall poor quality of life and well-being, including increased loneliness [16]. Addressing ageism within the healthcare setting needs to take priority as a significant lack of training leads to problems such as misdiagnosis and medication errors [17]. When older adults are not taken seriously concerning complaints they have about an issue, these concerns often go ignored, exacerbating a diminished quality of life and increasing the risk of hospitalization or death [18,19,20]. The lack of geriatric training is a barrier to improving older adult wellbeing and geriatric healthcare; thus, there is a need for university students in projected healthcare professions to receive positive experiences with older adults before entering the field [21]. There is a high demand for healthcare workers that are specialized in geriatric care, but a combination of low financial incentives, lack of educators, and most importantly a low student-centered initiative in the field has limited promotion and growth in healthcare fields [22,23]. Additionally, geriatric training is limited in professional programs and has unique circumstances to a number of healthcare specialties such as nursing, medicine, social work, occupational and physical therapy, pharmacy, and dentistry. These healthcare specialties are at their own individual discretion to determine what and how much geriatric training to include within their own respective programs [24].

From a wider view, according to the Centers for Disease Control and Prevention (CDC), the risk of dementia may increase by 50% when associated with social isolation [5]. The incidence of disease and poor health stemming from social isolation translate directly to increased expenditure for US taxpayers as government programs, on average, must spend $1608 more every year for each older adult with a limited social network as compared to those with good social connections, involving an estimated 14% of all older adults [14]. Altogether, this type of support results in Medicare costs exceeding $6.7 billion (about $21 per person in the US) annually [16], when such funding could be best utilized to provide interventions for the social isolation and loneliness of older adults. Within the last few years, there has been a dramatic increase in social isolation and loneliness within geriatric populations [25].

Having established the drastic impacts of COVID-19, the development of new programs and methods is necessary to appropriately care for isolated and lonely older adults, especially in long term care settings. A quality improvement project in nursing homes reveals that 68% of residents expressed some form of social isolation, loneliness, and depressive symptoms [5]. To address the significant burden that emanates from social isolation and loneliness, we posit that a targeted and structured focus on intergenerational human connectivity could increase social wellness and improve health outcomes.

There is a well-established body of literature on the importance of intergenerational relationships and programs. Intergenerational programs are designed to actively bring the younger generation and the older generation together with the purpose of improving both generations [26,27]. Intergenerational programs can benefit both generations through a combination of emotional, mental, physical, social, and sensory stimulation [26,28]. Intergenerational programs that are geriatric-centered pose the additional benefit of fostering positive attitudes towards older adults by exposing students to the realm of geriatrics, fostering potential interest in the field [29,30,31]. Although some studies regarding potential health benefits stemming from intergenerational programs can be found within the literature, there is a dearth of studies establishing them as nonpharmacological interventions in addressing social isolation and loneliness specifically. 

To address the significant burden that emanates from social isolation and loneliness, we posit that a unique approach that entails a targeted and structured intergenerational focus on human connectivity can increase social wellness and improve health outcomes. This project has strong clinical implications for health care providers utilizing community resources such as intergenerational programs to address social isolation and loneliness detected during clinical encounters. The use of nonpharmacological interventions in addressing these community health issues decreases preventable financial expenditures for nationwide healthcare systems while addressing systemic issues. 

## 2. Materials and Methods

The intervention, called Tellegacy, is a program that fostered a community service learning experience for university students, referred to as “Legacy Builders”, based on a tailored curriculum which seeks to decrease social isolation and loneliness in older adults in an effort to promote health and wellbeing. The Tellegacy program has been utilized by nonprofit organizations and over 30 older adult communities within the United States. This study focuses on the feasibility study phase of the Tellegacy intervention in the University of North Dakota, aiming to assess its practicality and viability for wider implementation in diverse older adult communities. The feasibility study is a crucial preliminary step in understanding whether Tellegacy can be effectively introduced and sustained among older adult populations, addressing key aspects such as recruitment, retention, acceptability, and resource requirements.

### 2.1. The Intervention 

Tellegacy is an intergenerational program of which mission is to engage older adults by keeping them connected with a social network while reinforcing their own individual experiences. In the program, university students, known as Legacy Builders, are connected weekly with older adults, known as Legacy Holders, one-on-one, through in-person, phone, or virtual visits once a week in an effort to provide a sense of hope and purpose for the older individual. Prior to participation in the program, Legacy Builders must engage in a synchronous online training session that instills core fundamentals regarding goal setting, mindfulness, growth mindset, reminiscence therapy, and practicing dialogue with an older adult. Once the Legacy Builder builds these skillsets, program staff match students with older adults who thereafter employ reminiscence therapy as the foundation for their weekly one-on-one sessions. At the conclusion of the program, Legacy Holders receive a Legacy Book, which is a book that contains visuals and summarizes stories shared by the older adult with the student. The Legacy Book serves to reinforce the stories that matter to the older adult. The implementation of Tellegacy is to become an intentional program that reinforces the existing social relationships of the older adult while also strengthening their personal drive and motivation for everyday activities.

### 2.2. Program Overview 

The program overview and management (Figure 1) consists of Older Adults/Student Recruitment, Student Training, Pre- and Post Assessments, Older Adult/Student Matching and Weekly Sessions, and Legacy Book creation. The Tellegacy Processes and Procedures Manual outlined the various steps involved in the intergenerational program between university students and older adults aimed at decreasing social isolation and loneliness.

### 2.3. Older Adult Recruitment 

Older adults were recruited from individuals living in-home or in assisted living facilities (ALFs) located throughout the Great Plains region of the United States. These facilities were both urban and rural, ranging from 30 to 200 rooms units that included independent and assisted living quarters. Most ALFs offered level 1 or 2 assistance and residents enrolled in either level were eligible to participate. 

Administrative leaders from Assisted Living Facilities were contacted by the PI to introduce the Tellegacy program and its logistics. A memorandum of understanding was created to outline the roles of the ALF staff, its older adult residents, and university students. A manual of operating procedures for Tellegacy was delivered to each participating ALF. Older adults were recruited by posters at the ALF, word of mouth by ALF staff and residents, as well as presentations at the ALF by the PI. 

Inclusion criteria included older adults, aged 65+, living in-home or within senior communities, willingness to meet with a university student for one-hour sessions, ability to consent for participation, non–terminally ill from cancer or end-stage chronic conditions, cognitively intact. The older adult exclusion criteria included terminal illness, hospice or palliative care, uncompensated aphasia/apraxia, non-English speaking, unwillingness to complete 4 sessions with students. The project informed the older adult about the project design, its goals, and the facilitation of older adult participants prior to implementation of the intervention. 

### 2.4. University Student Recruitment

In the Tellegacy feasibility study, recruitment focused on university students with desired majors in healthcare-related fields, such as Medicine, Public Health, Physical Therapy, and Nursing. The purpose of a diverse representation of majors can display interprofessional experiential education and reflect the interdisciplinary nature of the program. 

The onboarding process included the recruitment of students through channels such as Honors Programs, Greek Life, in-person speaking engagements, and STEM initiatives. A Google sign-up form was sent and once completed, students’ profiles and contact information were manually added to an Excel sheet under the folder name “Older Adult and Student Contact Information”. Consent forms were sent by the program assistant through email, requiring student signatures, and were stored in the team’s online REDcap and the Legacy Builders folder provided by the PI on Microsoft Teams. University students were recruited through posters, in-person meetings, e-boards, social media, classroom announcements, students participating through a course or unpaid internship, electronic university-wide mailing lists, or honors societies to fulfill community service hours. 

### 2.5. Student Inclusion/Exclusion Criteria

Student inclusion criteria included part time and full time undergraduate and graduate students that were willing to spend 3 h in training, in healthcare-related field or interest in a healthcare-related field, minimum 4 h of interaction with older adults and 4 h preparing a Legacy Book in addition to ~2 h of survey completion and participation in focus groups. Student exclusion criteria included non-enrolled university students, students on academic or disciplinary probation, high school students taking college level courses, or students enrolled exclusively in online university courses. 

### 2.6. Assessments 

The data collected for the older adults consisted of 3 pre- and post surveys and one demographic survey. For the older adults, these surveys included the standardized University of California Los Angeles (UCLA) Loneliness Scale, the Satisfaction of Life Scale, which quantified the psycho-cognitive status of individuals, and the PHQ-9 depression scale. For the Legacy Builder participant, a demographic survey, Ambivalent Ageism Questionnaire, and project satisfaction survey were completed. The data from the Tellegacy research project were combined with the pre- and post data in one data file (with a single ID for each person). This was implemented to allow for an evaluation of changes over time (e.g., the degree and direction for each person). 

Surveys and questionnaires were conducted using various scales. The UCLA Loneliness Scale (UCLA), Satisfaction of Life Scale (SOLS), and PHQ9 depression scale were used to assess the well-being of the older adults. The surveys were typically self-reported, completed through mail, or graduate student interviews. The demographic survey collected information regarding gender, age, veteran status, education level, income, financial distress, environmental distress, rural residency history, family size, and whether the participant was the first generation in college. The UCLA Loneliness Scale was used to measure loneliness and the Satisfaction of Life Scale assessed overall life satisfaction [32,33,34,35,36,37,38,39,40,41,42]. The PHQ9 depression scale was employed to evaluate depressive symptoms [43,44,45]. These scales were chosen for their sensitivity and specificity in capturing relevant aspects of the participants’ experiences. 

Similar evaluations were conducted with university students before and after their participation. The demographic survey collected information regarding gender, age, veteran status, education level, income, financial distress, environmental distress, rural residency history, family size, and whether the participant is the first generation in college, and a project satisfaction survey was administered online to assess their satisfaction with the program. The Ambivalent Ageism scale was also used to measure students’ attitudes toward aging, and it was administered online as well [46,47,48,49,50]. A project satisfaction survey was administered to assess the satisfaction levels of university students regarding their participation in the intergenerational program. The survey was designed to capture their overall satisfaction, perceived benefits, and any challenges faced during the program. The survey was administered online to ensure convenience and efficient data collection. 

### 2.7. Four-Week Rationale 

The decision to implement a short 4-week intervention phase for the Tellegacy program’s feasibility study was based on several strategic considerations. Firstly, it was implemented to efficiently gather initial data and insights while identifying potential challenges and areas for improvement. This approach allowed the program organizers to manage resources effectively, ensuring the feasibility study could be conducted within existing constraints. Given the potential sensitivity of the participants, a shorter intervention phase also offered flexibility for program adjustments and minimized participant burden. The decision on whether to continue or expand the program will depend on the findings of the feasibility study, with potential directions including program extension, introducing additional opportunities, and scaling-up based on positive outcomes and strategic objectives. Thus, the rationale behind the short feasibility study duration is to test the program’s viability in a practical setting, optimize resource utilization, and maintain flexibility in program adjustments while strategically assessing its potential for future expansion and enhancement.

### 2.8. Training Curriculum

Prior to becoming a Legacy Builder, students completed a Tellegacy curriculum and demonstrated the competencies sought for older adult reminiscence encounters. The curriculum revolves around Mindfulness/Gratitude, Growth Mindset, Goal setting/Visualization, and Reminiscence Therapy. Mindfulness and gratitude allow the older adult to reflect on what they appreciate in life. The curriculum emphasizes students understanding What Matters Most to the older adult. The student-guided goal-setting is older adult-centered, with examples such as “smiling more” or “having two meaningful conversations every week.” These small behavioral changes facilitated a growth mindset for the older adult. Reciprocally, students also participate in these goal setting and mindfulness exercises. The Legacy Builders prepared for the program through a 3 h intensive virtual training module. The following sections provide details regarding the Legacy Builder training curriculum:

Telehealth etiquette: Telehealth etiquette or “tele-presence” competencies include attention to distractors, audio quality, and video presence. Proper communication with older adults emphasized using a slow voice cadence with low speech frequency. 

Building Connection and Rapport: Legacy Builders learn about active listening and open-ended questioning of older adults [51]. Additionally, students learned key factors related to the Geriatric 4 M’s of What Matters, Mentation, Medication, and Mobility to enhance their sensitivity to older adult needs [52]. Through proper older adult listening and structured conversation, student volunteers developed trust with the older adult in an effort to facilitate free and open communication.

Goal setting and Visualization: Prior research has established goal-setting as a known method to build resilience and develop a sense of purpose with a given individual [20]. Through the training, students learned about employing goal visualization in their own lives as well as with their older adult partner.

Mindfulness: Students learned about mindfulness to build a sense of awareness and connectedness. This module instructed students to keep present and mindful not only of themselves but also of the Legacy Holder during the conversation. Being present in the moment with the older adult not only creates a more dynamic conversation but also builds a deeper sense of awareness and connectedness between the older adults and students [21,53,54]. 

Growth Mindset: When conversing with an older adult, the Legacy Builders learned to ask questions that encouraged a growth mindset. For example, when an older adult shared a difficult past experience, the Legacy Builder first appreciated the sharing of the older adult’s story and then redirected the conversation to what they mutually could learn from the experience.

Verbal and Spatial Reminiscence Therapy: The Legacy Builders became familiar with a broad spectrum of legacy questions that explored the older adult’s lifetime. The questions and answers fit into the overall framework of the structured conversation that took place during the Legacy Sessions. Within the post-session feedback form the Legacy Builders fills after every session, the memories mentioned by the Legacy Holder are detailed. These memories were aggregated to enrich the creation of the Legacy Book.

Appreciation and Sense of Awe: The structured conversation in each session developed a sense of awe and appreciation that is intended to reinforce the connectivity between the older adult and student. A sense of appreciation and awe not only enriches a conversation but also reinforces the value of the older adult sharing their personal stories with the Legacy Builder [55,56]. Students were taught to admire the lived experiences and wisdom shared by the older adult. This has potential to reaffirm the value of the older adult’s contributions to the conversations and possibly empower them in their sharing and connection.

Redirecting Conversation: The curriculum contains content made to familiarize Legacy Builders with the technique of redirection. While avoiding bias against the older adult’s story, students were encouraged to first acknowledge the older adult’s story and then direct the conversation in a way that was most edifying and constructive to further enrich the conversation. Senior volunteers of Tellegacy helped simulate moments with Legacy Builders during training where conversational redirection is desirable. 

### 2.9. Legacy Questions

The students were provided with a list of questions with life-related themes such as childhood and adulthood (see Table 1). Students were encouraged to ask follow-up questions to their scripted questions. The format of the questions was the same regardless of in-person or phone sessions. The questions contained a total of 145 questions to be asked over the course of the sessions with the Legacy Holders. These questions covered various aspects of the Legacy Holders’ lives, including their childhood, family, career, relationships, and experiences in different stages of life. For the purpose of the feasibility study, only 108 questions were provided for the students. To maintain an organic flow, students were not required to ask every question and were encouraged to share their own life stories as well with the older adult. They were designed to facilitate meaningful conversations and help students connect with and learn from the Legacy Holders.

#### 2.9.1. Matching Process

Matches between students and older adults were made based on preferences and availability of older adults, with upperclassmen given priority due to availability restrictions. Each Legacy Builder was matched with one Legacy Holder for the intervention. The program assistant sent match emails to inform the participants of their pairing. Throughout the program, there was ongoing communication and coordination between the program assistant, students, and older adults. Confirmation emails were sent to students with details of their meetings, and additional confirmations were sent 24 h prior to each session. A check-in system was implemented to track attendance and reschedule sessions if necessary. Post-session forms and surveys were administered to gather feedback and assess the progress of the program. 

#### 2.9.2. Weekly Sessions

The Legacy Builders used a series of scripted questions to ask the older adults during each session. After expectations were set at training for the four sessions, Legacy Builders used their newfound training to connect and optimally conduct the hour long visit with their paired older adult. Within each session, the flow of the conversation was not dictated. Instead, the Legacy Builders sought to create a positive environment to foster the new relationships. The Legacy Builders were tasked with building a rapport with their older adults through use of positive inflection and tones of conversation that hopefully fostered a sense of hope. The Legacy Builders encouraged their older adults to impart their wisdom and share past stories. The primary engagement entailed oral reminiscence as a therapeutic modality. Each session inculcated elements of goal setting, as mentioned earlier. 

After each weekly session between the dyads, the Legacy Builder completed a post-session form, summarizing the older adult’s stories and responses to reflective questions. Students had either of the following two options for data collection: Print out the questions for the particular session and take notes in a notebook in an effort to enhance conversation and connectivity with the Legacy Holder.Bring an electronic device capable of recording answers in a Word document containing the curriculum questions.

Midway through the program, contact was made with both older adults and students to ensure their satisfaction and address any concerns. Table 2 and Table 3 provide an overview. Post-session surveys were conducted, including the PHQ-9, Satisfaction of Life, UCLA-Loneliness Survey, and Tellegacy Satisfaction Survey. The book production process for older adults after the weekly intergenerational meetings involved converting their transcripts (post-session forms) into a Legacy Book.

#### 2.9.3. Legacy Books 

After each session, students completed a post-session form in which they were instructed to summarize the stories they heard from the older adults. They were also asked to reflect on how they may have felt impacted by the session. Once the students completed all the sessions, the PI and program assistant collected the completed post-session forms and performed further edits to create a Legacy Book. Dr. Holloway has a graduate degree in English and assisted in the creative and literary writing aspect of the Legacy Book creation. The PI worked with a graphic designer to create and ship the Legacy Books to the respective older adults within 30 days after the completed sessions and after all post-session forms were received. The content of the Legacy Books was curated to create a summary of the legacy of the older adult’s life while additionally highlighting how the students were positively impacted by the older adult sharing their experiences. 

## 3. Results

This feasibility study was designed to test the effect of the intervention (Tellegacy program) on older adults using pre- and post-test measurements. The intervention dyads encompassed older adults and university students in healthcare-related fields participating in the Tellegacy program. We compared the pre- and post-intervention measures. 

In the Tellegacy feasibility study, the majority of university student participants represented diverse academic majors, including Public Health, Electrical Engineering with desire to pursue healthcare, Biochemistry, Physical Therapy, Nursing, and a combination of majors including Public Health Education, MPH, and Pre-Medicine. This diverse representation of majors reflects the interdisciplinary nature of the program and its potential impact on students from various academic backgrounds.

In this particular project, Tellegacy connected university students with 11 older adults, one-on-one (Tellegacy dyads) in the northern Midwest area of the United States. The dyad meetings, called sessions, were 1 h virtual (conducted through Zoom or similar online meeting software), over the phone, or in-person encounters based on the older adult’s communication preferences. The research project was a study that examines the impact of intergenerational encounters on older adults. To this effect, the Legacy Builder and older adults met for 1 h, for a total of four sessions, once a week, and data were collected between January and May of 2023.

### 3.1. Data Collection

A total of eleven older adults participated in this feasibility study, with seven engaging in face-to-face meetings with University students in Grand Forks, North Dakota, while four interacted with students through weekly telephone sessions. The primary outcome measures, both pre- and post-intervention, remained consistent across the study. Older adults were evaluated using the Satisfaction of Life, Modified UCLA Loneliness, Preliminary Demographics, and PHQ9 Scale to assess their life satisfaction, loneliness levels, and depression levels. Meanwhile, students completed the Ambivalent Ageism scale online. Additionally, demographic information was gathered, encompassing variables such as Gender, Age, Veteran status, Education, Social class, Environment, Family Size, and First Generation in College status.

In preparation for the Tellegacy encounters between students and older adults, the study participants underwent a series of assessments administered by the students. These assessments included the UCLA Social Isolation and Loneliness Scale, the Satisfaction of Life Scale, and the PHQ9 Depression Scale. Subsequently, these assessments were repeated four weeks following the commencement of the Tellegacy encounters, serving as a pre- and post-intervention evaluation to gauge changes in the participants’ well-being over time in response to the feasibility study’s intervention.

### 3.2. Data Analysis

In the Tellegacy feasibility study, the demographic questionnaire results provide insights into the composition of the older adult participants, aligning with the focus on feasibility assessment. The majority of participants, accounting for 72.73% of the group, were female, with males constituting 27.27%. Age distribution was as follows: 36.36% were in the 90–96 age range, 27.27% fell into the 85–89 category, 18.18% were in the 80–84 range, and 18.18% were aged 65. In terms of educational background, 54.55% had a college education, 27.27% completed high school, and 18.18% had an 8th-grade education, as summarized in Table 4.

Regarding income, 63.64% of participants were classified as having medium income, while 36.36% were categorized as having low income. Some financial distress was reported by 18.18% of participants, and 9.09% reported experiencing environmental distress. A significant proportion, 72.73%, had past experience living in rural areas, with an equally high percentage, 72.73%, having a rural background. Currently, 45.45% resided in rural settings. Family sizes varied, with 54.55% having 6–7 members, 27.27% having 8–14 members, and 18.18% having 4–5 members. Moreover, 63.64% were the first generation in their family to attend college, while the children of 18.18% of the participants were the first and the details of 9.09% of the participants were unknown.

Descriptive statistics, including mean, standard deviation, and frequency, were computed for demographic variables and outcome measures as part of this feasibility study. Participants with missing data in the three primary measures (UCLA, SOLS, and PHQ-9) were excluded from the analysis. Missing data in covariates resulted in the exclusion of participants from the repeated measures design or any stratified analyses, particularly if the covariate with missing data was significant in the model. Outliers were identified in the three measures (UCLA, SOLS, and PHQ-9), with values more than three standard deviations from the mean being considered as outliers and subsequently excluded from the analysis. A similar procedure was applied to the analysis of student data, specifically with the Ambivalent Aging measure. To assess the relationship between self-reported outcomes (UCLA, SOLS, and PHQ-9) and psychological biomarkers, separate generalized linear models were conducted for each outcome, incorporating psychological biomarkers as predictor variables. Additional models included demographic variables as covariates. A generalized linear model was employed to ascertain any significant improvement in the Ambivalent Aging measure.

The university students consisted of undergraduate and graduate students. Table 4 summarizes the feasibility study’s demographics of the older adults. Despite the small sample size, we were impressed with the recruitment of individuals with rural backgrounds (72.7%). Consistent with expectations for residents of Assisted Living Facilities (ALF), nearly two-thirds (63.6%) reported a moderate income, while approximately one-fifth grew up in environments marked by financial or environmental distress.

#### Older Adult Assessment Results

To determine the minimum number of older adults required to detect a significant difference in our assessments, we analyzed the data results. The purpose of this process was to also determine how many participants would be needed within a larger potentially subsequent study. Average scores for UCLA, SOLS, and PHQ-9 were compared before and after the intervention using paired t-tests. Notably, UCLA scores increased by an average of 1.5 points, reaching statistical significance via both a paired t-test (*p* = 0.034) and a non-parametric Wilcoxon test (*p* = 0.036). Similar findings were observed in the other assessments, with SOLS significantly decreasing by 3.5 units (*p* = 0.038) and PHQ-9 increasing by 0.36 units (*p* = 0.001).

Power calculations were performed based on the mean differences and standard errors. Among the tests, only PHQ-9 exhibited sufficient power to detect a difference of 1.4 with the given sample size of 11 subjects. Power analyses were subsequently conducted to determine the necessary sample size for detecting differences within the observed range in this study. Table 4 shows what participant number would be needed for significant results in a larger study. Figure 2 provides an overview of the power analyses, where a power of 0.8 is indicated by solid lines and 0.9 by dashed lines. Notably, PHQ-9 (represented by green lines) demonstrated that a large sample size was not necessary to detect a difference of 1.4. For UCLA (blue lines), a sample size ranging from 20 to 38 subjects would be required to determine that mean differences in the range of 1 to 2 were not significant. SOLS (dark yellow lines) demanded the largest sample size, ranging from 15 to 55, to detect changes in the mean ranging from 3 to 4.

**Figure 2 ijerph-20-07094-f002:**
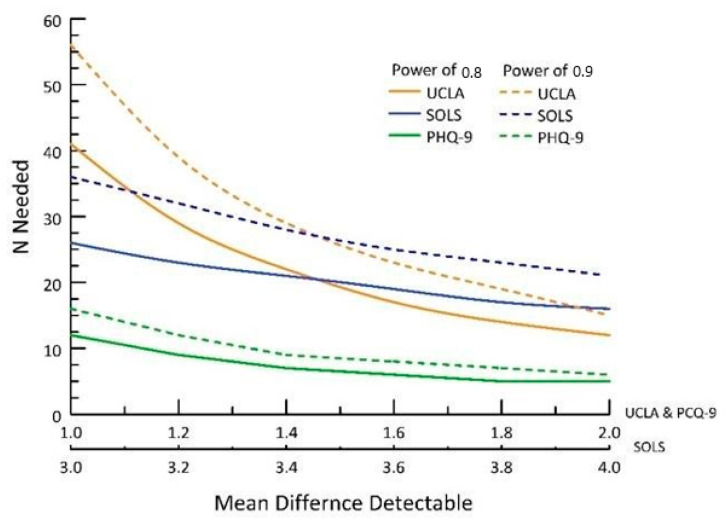
Power analyses of three measures.

**Table 4 ijerph-20-07094-t004:** No. of participants needed per scale.

Assessment Scale	No. Participants Needed (at Least)
UCLA Loneliness Scale	20–38
Satisfaction of Life Scale	15–55
PHQ-9 Depression Scale	12–15
Ambivalent Ageism Scale	18–25

Figure 3 shows the pre and post-Tellegacy results of eleven older adults, including the distribution of the score differences. Table 5 shows the values of the mean differences and their statistical assessment (albeit, feasibility data and small sample sizes are by definition indeterminant for statistical significance). UCLA scores decreased on average by 1.545 points (*p* = 0.034 using a paired t-test and *p* = 0.036 using nonparametric-9 decreased by 0.364 units (*p* =0.001).

### 3.3. Participant Retention

It is noteworthy that the feasibility study received a 100% participant retention rate, indicating that all enrolled individuals remained engaged throughout the study, without any dropouts or withdrawals. This underscores the robustness of the study design and the acceptability of the intervention among the participant group. Such a high retention rate suggests that the research methods and intervention procedures were well-suited to the needs and preferences of the target population, minimizing potential barriers or disengagement factors.

### 3.4. Participant Acceptability 

In line with the primary objective, the study also received a 100% acceptability rate among participants. This implies that all individuals involved in the study found the research procedures and the Tellegacy intervention itself highly acceptable and willingly participated. This level of acceptability reaffirms the feasibility of the intervention within the chosen demographic, indicating alignment between the Tellegacy program and participant preferences.

## 4. Discussion

### 4.1. Feasibility Implications for Older Adults

Adequate social connection remains a necessity for humanity as humans are a social species. Recognizing the fundamental human need for social connection, previous research has underscored the significant impact of social connections on both physical and mental well being [57]. Inadequate social relationships are known to contribute to the development of social isolation and loneliness. Within the context of this feasibility study, Tellegacy was designed with the aim of reducing self-reported isolation scores and enhancing various health metrics. These improvements in health, such as a heightened sense of purpose, have the potential to reduce the risk of conditions such as obesity, sedentariness, and sleep problems [58]. Notably, Figure 3 and Table 5, presented in the Results section, provide insights into the positive health enhancements observed among the Legacy Holders, aligning with the objectives of this feasibility assessment. Furthermore, within the context of a feasibility study, achieving 100% retention and 100% acceptability is a positive outcome. It suggests that the study design and intervention are not only feasible but also well-received by the participants. This bodes well for the broader implementation of the intervention as it demonstrates that it is both practical and in alignment with the preferences and needs of the target population.

**Table 5 ijerph-20-07094-t005:** Differences in means of UCLA, SOLS, and PHQ-9 results.

Measure	Mean Difference	Standard Error	*p*	Wilcoxen p	Power
UCLA	1.545	0.755	0.034	0.036	0.603
SOLS	−3.545	1.796	0.038	0.004	0.577
PHQ-9	1.364	0.338	0.001	0.005	0.982

### 4.2. Additional Screenings 

Surprisingly, despite the well-established influence of social wellness on physical health, usual clinical encounters rarely assess patient isolation and loneliness. Little evidence shows clinical research providing validated tools for the rapid assessment of these conditions. Tellegacy offers a prescriptive intervention to promote wellness while managing social isolation and loneliness. 

Nonetheless, the CDC and other professional groups clearly advocate for social wellbeing as a conduit to better health. This suggests that Tellegacy has enormous potential for providing an evidence-based intervention to either prevent or manage social isolation and loneliness. Within the scope of this study, these observations underscore the potential relevance and practicality of the Tellegacy pilot intervention study.

### 4.3. Reflections and Impact from the Legacy Builders

In the context of the program’s feasibility, it is essential to acknowledge the dynamics observed in the interactions between university students and older adults. While encounters with older faculty members or family members may partially bridge the intergenerational gap, a significant divide often exists between university students and older adults. This disconnect is exacerbated by students’ limited awareness of the challenges associated with social isolation among older adults, potentially attributed to by the segregation of older adults into retirement centers and long-term care communities. Additionally, language and cultural barriers may further isolate older adults, especially when interacting with community caregivers.

Through participation in Tellegacy, students reported gaining personal knowledge and experiencing self-actualization. Notably, they also recognized the program’s potential for fulfilling university or community service-based learning requirements. Future investigations could explore whether student interest in working with older adults increases while intrinsic biases against this demographic diminish as a result of their involvement in Tellegacy.

An area for improvement in future trials lies in streamlining data collection for students. Students found the process of submitting data in an Excel format cumbersome and expressed a preference for a Word Document version. Incorporating established data collection forms like Qualtrics or RedCap may enhance student accessibility and ease in data collection.

### 4.4. Limitations—Sampling

Participant self-selection into the Tellegacy program introduces the potential for selection bias, which may affect the generalizability of the findings. Social desirability bias could also be a factor as participants might provide responses perceived as more favorable or acceptable. 

The study’s duration was limited to a 4-week intervention period, which may not capture long-term effects adequately. Future investigations will consider longer intervention periods to evaluate sustained impact.

While this study lacked a control group, incorporating one in future Tellegacy trials will be crucial for establishing a baseline for program impact evaluation and determining effectiveness. This control group will facilitate comparisons with those who did not participate, offering valuable insights into the program’s effects in more detail and aiding in its refinement and enhancement. However, it is important to acknowledge that without random assignment and with a small sample size, systematic differences between the control and intervention groups may affect the validity of comparisons.

Despite these limitations and potential confounding variables, it is important to note that the results of this feasibility study remain valid as program staff maintained contact with participants to mitigate any potential burnout.

Additionally, the study raises questions about the impact of Tellegacy on different racial and ethnic groups. The current sample, consisting mostly of white, non-Hispanic, female, and rural older adults from the upper Midwest, reflects a homogenous population. Future efforts will seek to engage more diverse populations, including Indigenous communities, addressing potential disparities. Tools to monitor attrition rates, especially for populations of various disparities, although not observed in this study, are strongly considered for future research to account for any potential dropouts. To achieve this, allocated funds may be necessary to support access to underserved communities, ensuring greater inclusivity. The Tellegacy intervention’s continued impact will be bolstered by rigorous Legacy Builder training and consistency achieved through peer mentorship or faculty guidance.

The conceptual framework of Tellegacy, with its focus on mind–body interactions as individuals age, opens the door to future research on how social well-being contributes to healthy lifespans. This relationship prompts further exploration of how the Tellegacy intervention may impact physiological parameters such as functional status, biological aging, and healthy longevity.

Lastly, family members were allowed to participate in the study by providing photos or, with the older adult’s permission, attending weekly sessions; however, further studies can investigate the impact of family inclusion. Family members also had opportunities for additional geriatric education through the Department of Geriatrics at the University of North Dakota, highlighting the program’s potential to engage broader community networks.

### 4.5. Potential Advantage of a Feasible Intervention 

Tellegacy’s intergenerational intervention exhibits potential as a unique and promising feasibility study due to its multifaceted focus on mindset training, the enrichment of future healthcare staff, and the empowerment of geriatric populations. This approach holds the promise of equipping participants with essential listening skills, fostering interdisciplinary collaboration, and promoting unwavering values regarding health equity.

The feasibility study aims to challenge ageist mindsets among both university students and older adults, aiming to deepen their understanding of the challenges faced by the geriatric community while fostering empathy and compassion. It seeks to explore the feasibility of Tellegacy’s strategic aspects, emphasizing its mission to prepare the future healthcare workforce through immersive experiences and hands-on learning. Additionally, the study examines the feasibility of enhancing listening skills, recognizing their pivotal role in effective healthcare communication. Tellegacy’s interdisciplinary approach, which brings together students from various majors and older adults from diverse backgrounds, has the potential to enrich the learning experience for all involved.

### 4.6. Ethical Considerations in the Feasibility Study

In this feasibility study, ethical considerations have been a paramount concern. Informed consent was diligently obtained from all participants before their engagement in the study, ensuring their confidentiality and data protection. The study strictly adhered to ethical guidelines and regulations mandated by the institutional review board, demonstrating the commitment to upholding ethical standards throughout the feasibility assessment.

## 5. Conclusions

Previous research has highlighted the potential positive impact of reducing isolation and enhancing one’s sense of purpose on health outcomes [20,58,59,60,61]. These improved health outcomes likely stem from health promotion and personal wellness. In the context of this feasibility study, we aim to explore the feasibility of replicating these benefits for older adults. The program offers the advantage of weekly visits from intrinsically motivated university students in healthcare-related fields, potentially fostering social wellness through relationship building and connection. Furthermore, the feasibility study seeks to investigate whether the time allocated for older adults to reflect on their life experiences and their sense of contribution to the next generation might serve as a motivating factor, reinforcing their sense of purpose. Additionally, the study examines the feasibility of providing older adults with a Legacy Book that details their life experiences and stories, potentially enhancing their quality of life, sense of worth, and overall sense of purpose.

### 5.1. Social Isolation and Loneliness in the Context of Feasibility

The feasibility study acknowledges the critical issue of social isolation and loneliness within the aging population, particularly in light of the aftermath of the COVID-19 pandemic. The study recognizes that social distancing restrictions have exacerbated these issues, leading to adverse mental health outcomes among vulnerable populations [62]. Social isolation and loneliness have been linked to mental health challenges, including depression, anxiety, and stress, as well as physical health problems and a higher risk of early mortality [63]. Notably, the feasibility study aligns with previous findings that suggest a correlation between feelings of loneliness and the development of Alzheimer’s disease and depression, as well as between social isolation and poorer physical health, reduced cognition, diminished mobility, and increased difficulties in daily living [56,57,64]. Moreover, reduced frailty has been associated with decreased social isolation and loneliness [65].

Recognizing that this is both a healthcare and community issue that requires early intervention, the feasibility study aims to explore potential interventions to address social isolation and loneliness in older adults. While it acknowledges promising interventions with similar curricula, such as Mindfulness-Based Stress Reduction (MBSR) and Mindfulness-Based Cognitive Therapy (MBCT), the study emphasizes the need for further research into these interventions and their underlying mechanisms. These interventions have shown promise in addressing depression and improving mental health among older adults [66]. The feasibility study acknowledges the potential benefits of mental strength-based practices, which provide non-pharmacological approaches to addressing mental health challenges, including social isolation and loneliness [66,67,68]. These practices are designed to be flexible and accessible, making them suitable for older adults facing transitions in care, temporary stays in facilities, or permanent moves from independent living environments [69]. Notably, the feasibility study recognizes that mindset practices and mindfulness can be particularly useful for individuals with mobility issues, enhancing their overall well-being and self-care, especially when travel and other forms of self-care become challenging [30,70].

### 5.2. Feasibility Implications

It is crucial to recognize that the benefits of this feasibility study extend beyond the older adults involved. The potential impact on volunteer students or workers, particularly those pursuing careers in healthcare-related fields, is of profound significance. The training curriculum integrated into the Tellegacy program encompasses various modules, including vision, goal setting, self-efficacy, motivational interviewing, mindfulness, active listening, and conversation skills. Beyond its primary focus on assessing the feasibility of improving the lives of older adults, this program also seeks to evaluate how interactions aimed at learning from and understanding the older adult’s perspective might enhance the Legacy Builder’s comprehension of older generations and their own life journeys. Moreover, the feasibility study explores the program’s potential to provide students in healthcare fields with opportunities to fulfill required clinical or service hours within a positive and enriching learning environment.

### 5.3. Tellegacy’s Vision and Feasibility

The Tellegacy program aspires to not only address the issue of social isolation and loneliness but also educate the community about these concerns. It seeks to empower individuals to create action plans that combat these challenges through the power of connectedness and the acknowledgment of the intrinsic societal value held by older adults. Tellegacy advocates for the celebration of each older adult’s life and their active inclusion in their own narrative. Through further research, technology, and intergenerational efforts, the program seeks to decrease the feelings of isolation and loneliness in older adults and create opportunities to foster a sense of self-efficacy, connectedness, and hope within the aging population, thereby enabling happier and healthier people.

## Figures and Tables

**Figure 1 ijerph-20-07094-f001:**
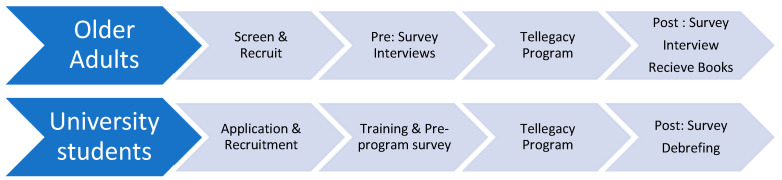
Timeline for the two groups of Tellegacy program participants.

**Figure 3 ijerph-20-07094-f003:**
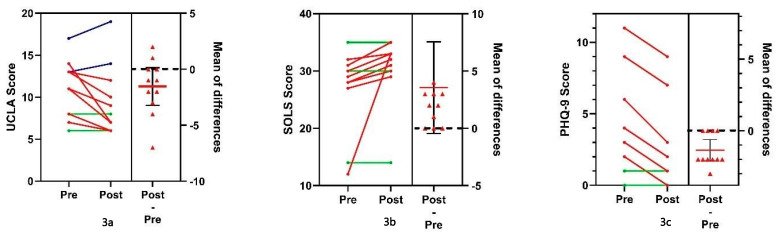
Pre and post scores with differences for UCLA, SOLS, and PHQ-9. Green lines signify no change. Red lines signify positive change. Dark blue lines signify negative change.

**Table 1 ijerph-20-07094-t001:** Weekly sessions and theme.

Weekly Session Number	Session Theme
Session 1	Introduction/Icebreaking Session/Childhood
Session 2	Childhood/Young Adult Life
Session 3	Young Adult Life/Adult Life (Spouse/Friend Relationships)
Session 4	Adult Life/Older Adult Life (Retirement/Grandchildren and/or lessons learned

**Table 2 ijerph-20-07094-t002:** Timeline of the intervention.

Week	Module
1	Student Training
2	Older Adult/Student Matching & Pre-assessment
3–12	Weekly sessions.
13–14	Post-assessment & Data Collection Begin Legacy Book creation

Note: Legacy Books sent to older adults within 30 days following last Legacy Holder/Builder session. Additional weeks allotted to supplement missed sessions.

**Table 3 ijerph-20-07094-t003:** Older adult preliminary demographics.

		*n*	%	Median
Gender	Male	3	27.27%	
	Female	8	72.73%	
Age	65	2	18.18%	86
	66–79	0		
	80–84	2	18.18%	
	85–89	3	27.27%	
	90–96	4	36.36%	
Veteran		2	18.18%	
Education	8th Grade	2	18.18%	
	High School	3	27.27%	
	College	6	54.55%	
Income	Low	4	36.36%	
	Medium	7	63.64%	
Financial Distress		2	18.18%	
Environmental Distress		1	9.09%	
Ever Lived in Rural		8	72.73%	
Rural Background		8	72.73%	
Current Rural		5	45.45%	
Family Size	4–5	2	18.18%	7
	6–7	6	54.55%	
	8–14	3	27.27%	
First Generation in College	Parent	1	9.09%	
	Theirs	7	63.64%	
	Children	2	18.18%	
	Unknown	1	9.09%	

## Data Availability

The data presented are available on request from the corresponding author. The data are not publicly available due to the small sample size, which potentially could compromise subject confidentiality.

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
