# Peer review of "Tellegacy: An Intergenerational Wellness and Health Promotion Project to Reduce Social Isolation and Loneliness in Older Adults: A Feasibility Study"

_ijerph, 2023, doi:10.3390/ijerph20237094_

Round 1

Reviewer 1 Report (New Reviewer)

Comments and Suggestions for Authors

In the following, I share my review for the manuscript titled "Tellegacy: An Intergenerational Wellness and Health Promotion Project to Reduce Social Isolation and Loneliness in Older Adults: A Pilot Study"

The topic presented in this pilot study is timely. The authors' focus on socio-emotional well-being of elders through interactions with young persons is an important contribution to today's intergenerational (IG) scholarship.
The authors developed and explored an IG program that was feasible in its implementation. However, the intervention seems brief. The Tellegacy program spanned eight weeks and engaged university students who interacted weekly with elders during four weeks. Additional weeks focused on training the Legacy Builders (university students) and creating a legacy book for the Legacy Holders (elders). Interactions between the students and elders took place during four weeks, which is a short intervention period. Brief IG interactions are most successful if participants have opportunities for sustained IG engagements.

It is unclear exactly how the interactions between students and elders took place. Where did the IG pairs meet? How did they interact during the four one hour legacy sessions?  

The authors need to more clearly discuss why they chose to implement a short intervention phase. Do they plan to continue the program and increase the intervention length? Or do they plan to implement other opportunities and approaches within the Tellegacy program for university students and elders to interact with each other?

The authors need to review the article for consistency of demographic data about the participants. I found references to three elders participating and eleven elders are featured in Table four. The authors mention one student taking the role of Legacy Builder and multiple students serving the roles of Legacy Builders. Were there 11 Legacy Builders and thus, students and elders were paired one to one? How were they paired?

For recruiting elders, the researchers discussed 65 and older as Legacy Holders. However, in Table four, the youngest age spans are 50 to 65 years old. Check for singular and plural references to students! Sometimes the authors talk about one student and sometimes they write about multiple students. Why were students recruited from multiple universities? Were the interactions conducted in person with pairs at multiple sites? If so, how many different sites? It is important to improve clarity regarding the participants and program implementation. Remember that the described program must be accessible for replication.

Check for the use of present, past, and future tenses throughout the manuscript! Be consistent!

Avoid long sentences. A paragraph must consist of more than one sentence (good to be at least three).  

Comments on the Quality of English Language

The use of English is adequate. For clarity of writing, please consider the suggestions in the previous section.

Author Response

Reviewer 1: 

It is unclear exactly how the interactions between students and elders took place. Where did the IG pairs meet?

Response: Additional information has been provided in the Methods section. Please review Section 2.7 Line 223 for further clarification.  

How did they interact during the four one-hour legacy sessions?  

Response: Additional information has been provided in the Methods Section 2.12.2 starting from line 370.  

The authors need to more clearly discuss why they chose to implement a short intervention phase. Do they plan to continue the program and increase the intervention length? Or do they plan to implement other opportunities and approaches within the Tellegacy program for university students and elders to interact with each other?

Response:  Additional information has been provided in the section entitled “Four-Week Rationale”; Section 2.9, Line 263

The authors need to review the article for consistency of demographic data about the participants. I found references to three elders participating and eleven elders are featured in Table four. The authors mention one student taking the role of Legacy Builder and multiple students serving the roles of Legacy Builders. Were there 11 Legacy Builders and thus, students and elders were paired one to one? How were they paired? For recruiting elders, the researchers discussed 65 and older as Legacy Holders. However, in Table four, the youngest age spans are 50 to 65 years old.

Check for singular and plural references to students! Sometimes the authors talk about one student and sometimes they write about multiple students. Why were students recruited from multiple universities? Were the interactions conducted in person with pairs at multiple sites? If so, how many different sites? It is important to improve clarity regarding the participants and program implementation. Remember that the described program must be accessible for replication. Check for the use of present, past, and future tenses throughout the manuscript! Be consistent! Avoid long sentences. A paragraph must consist of more than one sentence (good to be at least three).  

Response: Thank you; the manuscript has been updated to provide more accuracy, consistency, and clarification on the content mentioned.  

Reviewer 2 Report (Previous Reviewer 1)

Comments and Suggestions for Authors

The authors provided much greater detail of the Tellegacy program, which I appreciated. However, the authors did not take my suggestion to frame the paper primarily about the feasibility of the study. Instead, they performed a power analysis. I am not sure that a power analysis here is necessarily useful because 1)there is no comparison group 2)There are 11 participants (which means a lot of chance for variability)

Useful information could be recruitment (which you do mention), treatment fidelity, assessment processes, and treatment adherence (please see below).

https://www.ncbi.nlm.nih.gov/pmc/articles/PMC3081994/

Other, more minor notes:

It is not apparent to me how the new section regarding ageism fits into the paper. I can only see that this is relevant if 1) the university students are medical students and 2)a goal of the program is to reduce ageism among medical students. If that is the case, I would state that explicitly in the methods section. Otherwise, I would omit this paragraph. 

I think preliminary demographic  should go in the results rather than the methods.

The discussion section states "Finally, because of a lack of random assignment and a small sample size, the control group may differ systematically from the intervention group" I did not see any control group. I would state in the limitations section that there is no control group. It is also important to say why that is important. 

https://www.ncbi.nlm.nih.gov/pmc/articles/PMC3081994/

Author Response

Reviewer 2:

The authors provided much greater detail of the Tellegacy program, which I appreciated. However, the authors did not take my suggestion to frame the paper primarily about the feasibility of the study. Instead, they performed a power analysis. I am not sure that a power analysis here is necessarily useful because 1)there is no comparison group 2)There are 11 participants (which means a lot of chance for variability). 

Response: Thank you; the manuscript has been updated to be a feasibility study.  

Useful information could be recruitment (which you do mention), treatment fidelity, assessment processes, and treatment adherence (please see below).

https://www.ncbi.nlm.nih.gov/pmc/articles/PMC3081994/

 Response: Thank you; the manuscript has been updated to provide more accuracy and consistency. 

Other, more minor notes:

It is not apparent to me how the new section regarding ageism fits into the paper. I can only see that this is relevant if 1) the university students are medical students and 2)a goal of the program is to reduce ageism among medical students. If that is the case, I would state that explicitly in the methods section. Otherwise, I would omit this paragraph. 

Response: Yes, there is a strong emphasis on encouraging dyads between University students in healthcare-related fields with older adults.  

I think preliminary demographic should go in the results rather than the methods.

Response: Yes, the demographics have been moved to the results section of the manuscript.     

The discussion section states "Finally, because of a lack of random assignment and a small sample size, the control group may differ systematically from the intervention group" I did not see any control group. I would state in the limitations section that there is no control group. It is also important to say why that is important. 

Response: Thank you; We have made it clear that the feasibility study does not include a control group with why it is important to consider in the future. The manuscript has been updated to reflect these changes.

Reviewer 3 Report (Previous Reviewer 3)

Comments and Suggestions for Authors

Now the paper is ok.

Comments on the Quality of English Language

Minor editing is needed.

Author Response

Thank you for your review of our manuscript.  We have received your notes, "Now the paper is ok; minor editing is needed."    

We have updated the manuscript with minor editing to the draft and have uploaded the revised manuscript. 

Thank you

Reviewer 4 Report (New Reviewer)

Comments and Suggestions for Authors

It's interesting to read that interactions between seniors and students can improve the mental health of seniors. I think the structure of the paper is also good. There's only one problem. What kind of advantages does "Tellegacy", which is covered in this paper, have over other training methods? Please make the introduction and discussion a little richer so that readers can understand this. Once this work is completed, the paper is worthy of publication.

Author Response

Reviewer 4:

What kind of advantages does "Tellegacy", which is covered in this paper, have over other training methods? Please make the introduction and discussion a little richer so that readers can understand this. Once this work is completed, the paper is worthy of publication.

Response:  We’ve included an “Advantages” section in the Discussion; however, we desire to keep an objective and unbiased approach as much as possible. The mentioned sections have been revised for enrichment.   

Please make the introduction and discussion a little richer so that readers can understand this. Once this work is completed

Response:  We have revised and updated all sections of the manuscript and thank you for the review of our manuscript. 

Reviewer 5 Report (New Reviewer)

Comments and Suggestions for Authors

Dear author(s),

Thank you for sharing your work. May I please give you some feedback and I hope the feedback will be accepted in the spirit of collegiality. 

1. The introduction needs to be more cohesive. Although the necessary parts are there (ageism, loneliness, social isolation, economic costs), these parts need to be pulled together into a more cohesive whole. I am sure the the authors will be able to do this quite easily.

2. Please state which years the data were collected in, thank you.

3. Page 9 Line 378/379: the authors state 3 options but only 2 are noted in the text. Please correct.

4. What is C.8 in line 428 on page 10?

In it's present form with line throughs the manuscript is difficult to read. But, the reviewer can clearly see the results section and discussion section need to be reorganized to maintain clarity of content. May I please urge the author(s) to please reorganize these two sections? Thank you.

As COVID19 is now endemic, is it necessary to keep mentioning it in this paper? Would it not be better to remove all mention of COVID19 and focus on social isolation and loneliness as concepts of their own without connection to the pandemic? After all, even without the pandemic older adults were already facing loneliness and isolation. Thank you. 

Author Response

Reviewer 5:

  1. The introduction needs to be more cohesive. Although the necessary parts are there (ageism, loneliness, social isolation, economic costs), these parts need to be pulled together into a more cohesive whole. I am sure the authors will be able to do this quite easily.

Response:  Completed and updated

  1. Please state which years the data were collected in, thank you. 3. Page 9 Line 378/379: the authors state 3 options but only 2 are noted in the text. Please correct.4. What is C.8 in line 428 on page 10?

Response: Completed and updated

In it's present form with line throughs the manuscript is difficult to read. But, the reviewer can clearly see the results section and discussion section need to be reorganized to maintain clarity of content. May I please urge the author(s) to please reorganize these two sections? Thank you.

Response: We believe that this effect may have occurred for the reviewer due to viewing an older version of the manuscript with line throughs. A newer vision of the manuscript is available and optimized for viewing through Microsoft Word. 

As COVID19 is now endemic, is it necessary to keep mentioning it in this paper? Would it not be better to remove all mention of COVID19 and focus on social isolation and loneliness as concepts of their own without connection to the pandemic? After all, even without the pandemic older adults were already facing loneliness and isolation. Thank you. 

Response:  Provided that the Tellegacy Program started in light of the Pandemic, historically it would be accurate to mention the event; furthermore, the pandemic factually impacted healthcare fields and the epidemic of loneliness per US Surgeon General’s recent announcement, which has been further spun/shared to an exacerbating level (please see https://jamanetwork.com/journals/jama/article-abstract/2805292). Thus, we have decreased the amount of times COVID-19 is mentioned, as the authors agree that there is a need to lessen the emphasis on COVID. We have deleted a number of mentions related to the pandemic.  The authors agree that social isolation and loneliness have been a preexisting issue for older adults even before the pandemic.   

Round 2

Reviewer 2 Report (Previous Reviewer 1)

Comments and Suggestions for Authors

  I appreciate that the authors were responsive to most of my comments.

Although the framework of a feasibility study was woven into the introduction, the rest of the paper read like an evaluation of the program. The authors could be more consistent throughout in the way the results and discussion section are framed. 

Also, the authors should explicitly state in the methods section (not just the response to my comment) that most of the students were training to be in healthcare fields.  

Author Response

Reviewer 2: 

I appreciate that the authors were responsive to most of my comments.

Although the framework of a feasibility study was woven into the introduction, the rest of the paper read like an evaluation of the program. The authors could be more consistent throughout in the way the results and discussion section are framed. 

Response:

The framework of the feasibility study has been woven throughout the rest of the paper, including the results and discussion sections.  The structure of the results and discussion sections have been adjusted to follow a clearer and more consistent format.  For example, we have considered the tone in which results and discussion are presented, have made adjustments throughout and we considered use of headings and/or subheadings, including the explanation of findings to follow a more consistent feasibility study related theme.

Reviewer 2: 

Also, the authors should explicitly state in the methods section (not just the response to my comment) that most of the students were training to be in healthcare fields.  

Response:

Although the authors have previously placed students’ majors in the Methods section 2.5, line 187, the authors have more explicitly added the desired content in the recruitment section as well. In the Methods Section, 2.4 University Student Recruitment, content has been added, stating, “In the Tellegacy feasibility study, the majority of university student participants represented diverse academic majors, including Public Health, Electrical Engineering with desire to pursue healthcare, Biochemistry, Physical Therapy, Nursing, and a combination of majors including Public Health Education, MPH, and Pre-Medicine.”

This manuscript is a resubmission of an earlier submission. The following is a list of the peer review reports and author responses from that submission.

Round 1

Reviewer 1 Report

Comments and Suggestions for Authors

This paper presents a novel idea- to develop an intergenerational program to decrease loneliness and social isolation among older adults. The program in creative and has  potential to improve the wellbeing of older adults. However, there are aspects of the study design that need additional consideration.

Main comments:

1.  Is there a way to repeat this pilot with other university students? I think if you include more students and older adult pairs, you will get more information about intervention feasibility and acceptability.

2. The main contribution of this paper is to introduce the intervention and to describe feasibility/acceptability.  I do not think the results from a sample size of three individuals without a control group can be meaningfully interpreted and don't think presenting the results of the depression/loneliness scales at all is that meaningful. 

3. More information can be given about recruitment of both the older adults and the students.

4. Was any work done to examine the acceptability of the intervention to older adults?

5 Figure 2 needs to be more thoroughly explained and referenced in text.

6. The paper does a good job of explaining the training the legacy builder receives. I am a little confused regarding how and when the book is created. Also, what does it look like and what does it contain?

7. Information about the burden of the forms to the legacy builders is useful. However, I don't quite understand the difference between the standard forms and the Word document. I also don't know why writing out the text in the participants' own notebook is very helpful.

Author Response

Please check with the attachment.

Reviewer 2 Report

Comments and Suggestions for Authors

INTRODUCTION

The introduction needs improvement; there is no academic structure of the paragraphs. Three elements are essential here: What is known, what is unknown about the research question, and why the study was needed to carry out. The paragraphs must have an academic structure with an opening sentence, support sentences, and closing statement (sandwich structure).

It is important to reformulate the second paragraph "what is not known about the topic" in which the gaps or limitations that exist in the literature about the research question should be described. It is essential to describe the methodological deficiencies in previous studies, for example, unmeasured confounding, uncontrolled confounding, small sample size, lack of statistical power, studies without sufficient biostatistical rigor, among others.

METHODS

The authors should specifically mention their research design. In addition, it lacks basic data such as population, sample and sampling performed. Nor is a section describing the instruments used, including information on their psychometric properties (reliability and validity) reported. In addition, there is no detailed description of the statistical analysis used for the research.

RESULTS

The statistical analyses are not appropriate and are not well described in the methodology section. The authors should substantially improve the presentation and rigor of their results.

DISCUSSION

The authors need to improve their discussion.

In limitations, authors should add information bias, mentioning if there are any confounding variables that have not been measured in their research. Additionally, discuss the generalisability (external validity) of the study results. In addition, in the limitations described, it is not enough to mention each one of them, but they should mention what was the potential solution to deal with these limitations and explain why their presence does not invalidate their study results.

Additionally, I suggest strengthening your discussion with the redaction of an "Implications of findings in mental health policy" paragraph.

I suggest that this research be reformulated and probably presented in a letter to the editor. Otherwise, I suggest that a larger sample can be obtained in your research to communicate more rigorous scientific findings.  I do not recommend the publication of this manuscript in its current version, since it has serious methodological problems and requires a rethinking of the statistical analysis and a thorough revision and reformulation of its discussion.

Author Response

Please check with the attachment.

Reviewer 3 Report

Comments and Suggestions for Authors

This is an interesting study. 

In the introduction the authors should include more relevant research findings. Thy could add useful information about research and theories concerning their topic coming from a recent review: Tragantzopoulou, P., & Giannouli, V. (2021). Social isolation and loneliness in old age: Exploring their role in mental and physical health. Psychiatriki32(1), 59-66.

In addition to the above, a major methodological problem is that it is necessary to provide data coming from more participants. The N sample size is too small both for quantitative as well as qualitative research.

A last point to consider is the discussion. More references are needed in order to support your statements.

Overall an interesting preliminary study.

I

Author Response

Please check with the attachment.

Reviewer 4 Report

Comments and Suggestions for Authors

Thank you for the paper, “Tellegacy: Reflections from the Development of An Intergenerational Wellness and Health Promotion Project to Reduce Social Isolation and Loneliness in Older Adults: A Pilot Study.” This is a nicely done pilot study, and the results will be helpful for future study design. I have some minor comments:

  1. In Table 1, can the authors explain in table notes why there are two “0M” on each row?

  2. Figure 3. It may be more interesting to present the results that are colored by each older adult. Because the shape of the line itself already shows whether the scores improved, decreased, or stayed the same. Also, later in the results/discussion, the authors highlighted one particular older adult that scored a lower score in the Post UCLA Loneliness scale. So it will make more sense also to show the score changes by individuals in Figure 3.

  3. The authors may consider adding more discussion on the improvable characteristics of the program. 

For example, “The student was asked about her experience in training. She felt the video modules and the PDF on preparing for the one-on-one sessions prepared her well for the sessions, but she felt that the post-session forms, which provided all of the questions she needed to ask the older adults, were in a format difficult or too time consuming for her to fill them out with adequate time.” Any comments or suggestions on how to improve that? Specifically, future studies with a larger sample size warrant a more convenient, or user-friendly study design.

Author Response

Please check with the attachment.
